# Anaerobic and Aerobic Metabolic Capacities Contributing to Yo-Yo Intermittent Recovery Level 2 Test Performance in Australian Rules Footballers

**DOI:** 10.3390/sports12090236

**Published:** 2024-08-30

**Authors:** Mitchell Mooney, Ryan Worn, Matt Spencer, Brendan J. O’Brien

**Affiliations:** 1Faculty of Health, Federation University Australia, Mt. Helen, VIC 3357, Australia; mitchell.mooney@athletics.org.au (M.M.);; 2School of Behavioural and Health Sciences, Australian Catholic University, Melbourne, VIC 3605, Australia; 3Department of Sport Science and Physical Education, University of Agder, 4605 Kristiansand, Norway

**Keywords:** maximal aerobic speed, aerobic power, running economy, maximal accumulated oxygen deficit

## Abstract

This study aimed to identify the aerobic and anaerobic metabolic performance capacities contributing to Yo-Yo Intermittent Recovery level 2 (Yo-Yo IR2) test performance. Nineteen recreational Australian footballers completed a Yo-Yo IR2 test, and on another day a treadmill peak oxygen uptake (VO_2peak_) and maximal accumulated oxygen deficit test in a randomised counter-balanced order. The maximal accumulated oxygen deficit (MAOD) procedures included 5 × 5 min sub-maximal continuous runs at progressively higher speeds whilst VO_2_ was recorded; thereafter, speed was incrementally increased to elicit VO_2peak_. After 35 min of rest, participants ran at a speed equivalent to 115% of VO_2peak_ until exhaustion, at which point expired air was collected to determine maximal accumulated oxygen deficit. Relationships between variables were assessed using Pearson’s correlation and partial correlations. Maximum aerobic speed, relative intensity, and VO_2peak_ were significantly correlated with Yo-Yo IR2 performance. High Yo-Yo IR2 performers also had higher MAS, relative intensity, and VO_2peak_ levels. However, when higher maximum aerobic speed, relative intensity, and VO_2peak_ were controlled for each other and analysed independently, neither maximal aerobic speed nor VO_2peak_ correlated with Yo-Yo IR2 performance. Yo-Yo IR2 performance is the result of a complex interaction between several variables. Training programs should primarily focus on improving VO_2peak,_ maximal aerobic speed, and relative intensity to optimize Yo-Yo IR2 test performance.

## 1. Introduction

Australian Rules football requires athletes to perform numerous high-intensity efforts with short recovery [1]. Like other team sports, Australian Rules football coaches and physical trainers perform several field tests to identify the physiological and metabolic capabilities of athletes to cope with the repeated high-intensity demands of a match [2,3,4]. The Yo-Yo Intermittent Recovery level 2 test (Yo-Yo IR2) is a field test that strongly correlates with the high-speed locomotion in Australian Rules football and soccer [5,6].

One of the strengths of the Yo-Yo IR2 test in predicting Australian Rules football and soccer repeated-sprint performance is that it challenges a variety of physiological systems that interact in a similar way to match play in many team sports [7,8]. Further, it has been suggested that tests such as the Yo-Yo could be included as part of multi-measurement criteria for player selection, especially when it is challenging to discriminate sport-specific fitness between players [9]. Likewise, selected state-level Australian Football League (AFL) women players have been observed to perform better in the Yo-Yo IR1 test compared to their non-selected counterparts [10]. Differences exist between the Yo-Yo IR1 and IR2, demonstrating that Yo-Yo variations may be discriminating factors in player selection. 

Given the demonstrated links between Yo-Yo IR2 performance and team sport exercise intensity, as well as possible impacts on player selection, team sport training programs may attempt to measure the success of training interventions using this test (8). However, to improve Yo-Yo IR2 performance, knowledge of the relative influence of the aerobic and anaerobic metabolic capacities underlying its performance is essential to optimize a specific and periodized training regime. Aerobic metabolic capacity contributes substantially to Yo-Yo IR2 test performance as the test elicits maximum heart rate [11]. Yoshida and Watari [12] showed that distance runners with a high peak oxygen uptake (VO_2peak_) can resynthesize phosphate creatine more quickly than runners with lower VO_2_. In theory, quicker phosphate creatine resynthesis should enhance repeated high-intensity efforts. However, Rampinini et al. [13] showed that VO_2peak_ only moderately correlates with Yo-Yo IR2 (r = 0.47). Additionally, VO_2peak_ is not a strong discriminator of endurance sports performance in heterogenous samples, and other aerobic performance qualities such as maximal aerobic speed (MAS) and running economy can be more predictive of endurance performance [14]. Olsen et al. [15] recently showed that maximal aerobic speed correlates strongly with repeated-sprint ability in soccer players. The relationship of maximal aerobic speed and running economy to Yo-Yo IR2 performance has not been previously investigated. 

Yo-Yo IR2 test performance has large correlations with post-test blood lactate, H^+^ accumulation, and rate of lactate accumulation [7]. Consequently, a high anaerobic capacity appears to be an important determinant of Yo-Yo IR2 performance, as maximal Yo-Yo IR2 performance requires a sustained contribution from anaerobic glycolysis. 

As the influence of aerobic metabolic performance measures on Yo-Yo IR2 performance is unclear and the relationship of anaerobic capacity to Yo-Yo IR2 performance has not been previously investigated, this study aims to determine how these metabolic qualities correlate with Yo-Yo IR2 performance. The hypothesis tested is that aerobic metabolic performance and anaerobic capacity measures are correlated with Yo-Yo IR2 performance. 

## 2. Materials and Methods

### 2.1. Subjects

Nineteen male regional team sport athletes (Australian Rules footballers) from various teams volunteered to participate in this study in response to advertising flyers placed at football clubs and the University. Participants were included if they were male and had been actively participating in non-professional Australian football. Participants were asked to maintain a consistent diet in the 24 h leading into the testing days, including the discouragement of the use of caffeine. The participants were competing in various regional competitions of a similar standard (non-professional), completed at least 5 h of Australian football-specific activity per week (training and matches), and had a mean (±SD) stature of 180 ± 5 cm, mass of 78.0 ± 6.0 kg, and age of 20.7 ± 1.6 years. Ethical approval was obtained from the University Human Research Ethics Committee (project approval number A11-002) and the study complied with the Declaration of Helsinki’s ethical guidelines. The risks were presented to each participant in writing and verbally prior to the study. Written informed consent was then collected from each participant.

### 2.2. Procedures

The study had a correlational design, with subjects completing the protocols in a randomized, counter-balanced order.

Each participant was required to complete the Yo-Yo IR2 test and metabolic capacity tests on separates days within 2 weeks of each other. The Yo-Yo IR2 test was completed on a stable indoor surface using procedures established previously [7]. Participants were familiarized with the testing protocols of Yo-Yo IR2 by participating in a sub-maximal Yo-Yo IR2 test up to 48 h prior to testing without exhaustion. These procedures have been found to be reliable and applicable to team sports such as soccer and Australian Rules football [5,7]. The metabolic tests were performed in an exercise physiology laboratory. The aerobic metabolic performance capabilities assessed were peak oxygen uptake, maximal aerobic speed, and relative intensity (running economy). The anaerobic metabolic capacity was determined by the maximal accumulated oxygen method (MAOD) [16]. The MAOD protocols were familiarized to the participants with a non-exhaustive practice run. All participants indicated that they had used the laboratory equipment before for similar testing protocols.

Participants were in pre-season and not competing at the time of testing; testing was performed on a day off from training for all participants at a time of their convenience. However, the Yo-Yo IR2 tests and metabolic tests were performed at a similar time of day for each individual. Participants were required to perform 2 × 20 m shuttles in time to an audio beep of progressively increasing intensity with 10 s of rest between shuttles, in which the participant jogged around a cone situated 5 m past the start/finish line. Each participant received a warning if they did not make the shuttle in time; upon missing it a second time, they were eliminated from the test. The total distance covered in meters during the test was deemed their Yo-Yo IR2 score in accordance with Krustrup et al. [7].

The MAOD protocol was conducted in a laboratory on a motorized treadmill ergometer at a 1% gradient (T200, Cosmed, Rome, Italy) to replicate the energy cost of over-ground running [17]. The participants ran at five progressively faster speeds for 5 min at each speed (8, 9.5, 11, 12.5, and 14 km·h^−1^) without a break between each speed for 25 min of accumulated running time. Before they ran, participants were fitted to a two-way breathing apparatus (Hans Rudolph, Shawnee, KS, USA) through which expired air was analysed by an online metabolic system (Moxus, AEI Technologies, Bastrop, TX, USA) previously calibrated with gases of known concentration (20.9% and 16% O_2_ and 0.3% and 4% CO_2_) and volume (3 L syringe). The volume of oxygen consumption was recorded every 30 s and averaged for the final two min of each 5 min period. Continuing without rest from the final submaximal run, the intensity was increased by 1 km·h^−1^ every minute until ventilatory exhaustion and VO_2_ plateau to obtain VO_2peak_. VO_2peak_ was determined as the highest value recorded in 30 s intervals and reflects the highest aerobic metabolic rate of the individual. Submaximal relative intensity was calculated to be the average VO_2_ (mL·kg^−1^·min^−1^) during the final two min of the 12.5 km·h^−1^ stage, presented as a percentage of VO_2peak_. Running economy was determined as the actual steady state VO_2_ (mL·kg^−1^·min^−1^) during the 12.5 km·h^−1^ stage.

The mean VO_2_ values for the final two minutes of each of the five submaximal run speeds were submitted to a linear regression model to determine the supra-maximal treadmill speed required at 115% of VO_2peak_. The coefficient was very high, ranging from r^2^ = 0.93 to 0.99, similar to prior work (16). The VO_2peak_ of each participant was substituted into the regression equation to determine their predicted maximal aerobic speed (MAS), defined as the speed corresponding to VO_2peak._ Participants were given 35 min of passive rest before completing the supra-maximal run to exhaustion; they were encouraged to drink fluids ad libitum and restricted to ingesting less than 20 g of carbohydrate to avoid possible gastrointestinal distress. Participants ran at a speed calculated to reflect 115% of VO_2peak_ on a 1% gradient until exhaustion. Participants waited until the treadmill reached the prescribed speed, then lowered themselves onto the treadmill belt. Data collection began when the participant removed their hands from the rails of the treadmill. Collection ceased once the participant indicated they could no longer maintain the speed either by pressing a stop button on the side of the treadmill or by a non-verbal signal to the test administrators.

The participants were fitted with a two-way breathing apparatus, with expired air being collected directly by Douglas bags and analysed using first principles [16]. The expired air was pumped into a metabolic system to determine the O_2_ and CO_2_ concentrations at a rate of 3 L per min; the extraction was timed, and the residual volume was included in the calculations. Expired air was then extracted manually by a 2 L syringe until the bags were empty to determine the total volume of air. The volume of air was then standardized for pressure, water content, temperature, and N_2_ content. VO_2_ was then calculated as below (Equation (1)). VO_2_I was calculated using the Haldane transformation, whilst VO_2_E was calculated from the percentage of O_2_ content multiplied by the standardized volume.
VO_2_ = VO_2_I − VO_2_E (1)

VO_2_ = Oxygen uptake.VO_2_I = Volume of oxygen inspired.VO_2_E = Volume of oxygen expired.

MAOD was then calculated as the predicted VO_2_ subtracted from the actual VO_2_, as previously described [16,18,19,20,21]. The duration of the supra-maximal run was also recorded. Each participant was scheduled for an hour in the laboratory to complete testing.

### 2.3. Statistical Analyses

Prior to the study, a power calculation was performed to determine the sample size required to detect an r of 0.6 with a power of 0.8 and a statistical significance of 0.05 in a two-tailed test. Nineteen participants were identified and recruited. Following the testing procedures, all data underwent statistical analyses using SPSS (version 26; IBM, Armonk, NY, USA). Normality was assessed using the Shapiro–Wilk test and via the visual inspection of histograms and Q-Q plots. Descriptive statistics (mean and standard deviation) were calculated for each variable. The data were separated into high (*n* = 9) and low (*n* = 10) Yo-Yo IR2 scores using the median split technique. All variables were then analysed for differences between high and low Yo-Yo IR2 groups using a one-way ANOVA. Partial correlations were also used to isolate the contribution of variables independent of other associated variables (found in the correlation matrix) [22]. The significance level for all statistical tests was set at *p* < 0.05.

## 3. Results

The Yo-Yo IR2 test revealed the participants’ mean (±SD) distance to be 689 ± 155 m, ranging between 440 m and 920 m. Furthermore the treadmill protocol revealed that the participants had a mean VO_2peak_ of 56.5 ± 5.9 mL·kg^−1^·min^−1^, a relative intensity of 84.1 ± 7.9%, a VO_2peak_ of 12.5 km·h^−1^, a running economy of 12.5 km·h^−1^ 47.3 ± 3.7 mL·kg^−1^·min^−1^, a predicted MAS of 15.0 ± 1.4 km·h^−1^, an MAOD of 13.7 ± 6.0 mL O_2_ eq·kg^−1^·min^−1^, and time to exhaustion at 115% MAS of 181.5 ± 53.1 s. A correlation matrix between the measured variables is also presented in Table 1. These results show significant relationships between Yo-Yo IR2 performance and VO_2peak_ (r = 0.62, *p* = 0.005), relative intensity (% VO_2peak_ at 12.5 km·h^−1^) (r = −0.72, *p* = 0.001), and predicted MAS (r = 0.70, *p* = 0.001). However, no significant relationship was revealed between running economy and Yo-Yo IR2 performance (r = 0.05, *p* = 0.86). When separated into high and low Yo-Yo IR2 groups, the mean Yo-Yo IR2 performance was 738 ± 102 m (ranging 640–920 m) for the high group and 492 ± 62 m (ranging 440–600 m) for the low group. There was a significant difference in VO_2peak_, relative intensity, and predicted MAS between the high and low Yo-Yo IR2 groups (Table 2).

Figure 1 displays a partial correlation analysis revealing that when controlled for VO_2peak_ and relative intensity, predicted MAS is not significantly related to Yo-Yo IR2 performance (r = 0.28, *p* = 0.28). However, when predicted MAS was controlled, VO_2peak_ and Yo-Yo IR2 performance showed a trend towards a relationship (r = 0.46, *p* = 0.095).

## 4. Discussion

This study aimed to identify the aerobic and anaerobic metabolic and performance capacities contributing to Yo-Yo Intermittent Recovery level 2 (Yo-Yo IR2) test performance. The results revealed that predicted maximal aerobic speed (MAS) and relative intensity had the strongest associations with Yo-Yo IR2 performance. Additionally, when participants were grouped based on their Yo-Yo IR2 scores, significant differences in MAS, relative intensity, and VO_2peak_ were observed. Further analyses uncovered a series of direct and indirect interactions between all measured variables, which can be used to identify areas of weakness in an athlete’s physical performance in sports where the Yo-Yo IR2 test is a valid measure. Overall, the findings highlight the intricate interplay between endurance qualities and Yo-Yo IR2 performance and suggest that a comprehensive approach is required to optimize athletic performance in such sports. Additionally, there was a significant correlation between Yo-Yo IR2 performance and VO_2peak_.

Improvements in VO_2peak_ have been shown to be coupled with improvements in Yo-Yo IR1 test performance and other repeated-sprint ability tests in previous training studies [23,24,25,26]. However, it is not clear whether the differences in this study were due to VO_2peak_ alone or due to the accumulation of effects of MAS and running economy. The results showed that running economy and maximal accumulated oxygen deficit (MAOD) showed indirect associations with Yo-Yo IR2 by correlating with VO_2peak_ and relative intensity.

It was also observed that the high Yo-Yo IR2 group had a greater VO_2peak_ but not great MAOD levels, whereas previous studies have shown that Yo-Yo IR2 has been moderately correlated with other anaerobic qualities such as agility and 5 and 10 m sprint times [27]. This finding highlights the clear interaction of physiological abilities that contribute to Yo-Yo IR2 performance.

Identifying the key interactions of physiological abilities that contribute to Yo-Yo IR2 performance can assist coaches in understanding strengths and weaknesses in individual athletes and enable them to identify improvement areas for athletes. The data showed that two participants obtained the same Yo-Yo IR2 score, but one had a slightly higher predicted MAS (16.4 km·h^−1^ vs. 16.0 km·h^−1^, ~2% difference) and a much lower VO_2peak_ (56.5 vs. 62.3 mL·kg^−1^·min^−1^, ~9% difference), indicating that the quality that has the greatest improvement potential may differ between individuals. This suggests that practitioners are unlikely to be able to infer individuals’ specific physiological traits from the Yo-Yo IR2 test reliably. This can likely be explained by the nature of the Yo-Yo IR2 test and the wide array of physiological traits that may affect its performance. Indeed, in other studies, strong correlations have been demonstrated between VO_2max,_ blood lactate accumulation, and phosphate creatine degradation and Yo-Yo IR2 test performance in male professional soccer players [8]. Strong correlations between agility [28] and power and acceleration in both senior and youth soccer players [29] and the Yo-Yo IRT1 test have also been reported. The wide array of potential correlates to Yo-Yo performance explains why it is difficult for any measure to relate to performance in all athletes. 

It is important to note that although the current study measured aerobic power and anaerobic capacity, there could be a contribution of other unmeasured physical capacities such as leg power, speed, agility, and acceleration. Some of these qualities have been previously tested in a similar protocol to the Yo-Yo IR2 test [27], indicating that agility and 5 m and 10 m sprint times are moderately correlated with the Yo-Yo IR2 test. Muscle phosphate creatine levels decrease after the completion of the Yo-Yo IR2 test [30]. Further research should focus on identifying the relative contribution of these qualities to the Yo-Yo IR2 test for a more comprehensive picture of the physiological requirements of the Yo-Yo IR2 test and to increase the knowledge regarding the strengths and weaknesses of individual athletes.

While these findings provide insight into the relationships between underlying physiological measures of aerobic and anaerobic capability, there are limitations to this protocol that are essential to acknowledge. Firstly, the participants do not represent the entire population of Australian footballers, and thus no predictive power can be provided for Yo-Yo IR2 based on these findings. Secondly, this study protocol did not include lactate testing, which may have provided an additional window into anaerobic metabolic contributions. Finally, we do not know if the Yo-Yo IR2 test is able to discriminate field running performance during matches.

Despite the above limitations, this study reveals that the most significant factors contributing to Yo-Yo IR2 performance were VO_2peak_, predicted MAS, and relative intensity (%VO_2peak_ at submaximal speed). However, none of these factors showed a significant correlation with Yo-Yo IR2 performance when examined independently. This suggests that Yo-Yo IR2 performance is influenced by multiple factors that interact with one another, and that it may be unrealistic to use this test alone to infer specific physiological characteristics at the individual level.

## 5. Conclusions

This study aimed to determine the aerobic and anaerobic metabolic performance capacities contributing to Yo-Yo Intermittent Recovery level 2 performance. The novel analysis of the physiological contributors to Yo-Yo IR2 performance identified that the most dominant interacting factors influencing Yo-Yo IR2 performance were peak oxygen uptake, predicted maximal aerobic speed, and relative intensity. Importantly, independently none of these variables showed a significant correlation with Yo-Yo IR2 performance. Therefore, Yo-Yo IR2 performance requires contribution from multiple interacting factors that can be monitored to determine the physical strengths and weaknesses of individual athletes. Therefore, inferring specific physiological characteristics from a Yo-Yo IR2 test in a precise or reliable manor may be unrealistic at the individual level.

## 6. Practical Applications

Repeated high-intensity running is an essential component of Australian Rules football. The Yo-Yo IR2 test is used in Australian Rules football training to determine footballers’ capability to perform the repeated high-intensity efforts required during competitive matches. The most likely metabolic performance measures to impact Yo-Yo IR2 are VO_2peak_, predicted maximal aerobic speed, and relative intensity; however, anaerobic metabolic performance capacities may also impact on performance. Therefore, training should focus on both aerobic and anaerobic metabolic qualities to improve Yo-Yo IR2 test performance.

## Figures and Tables

**Figure 1 sports-12-00236-f001:**
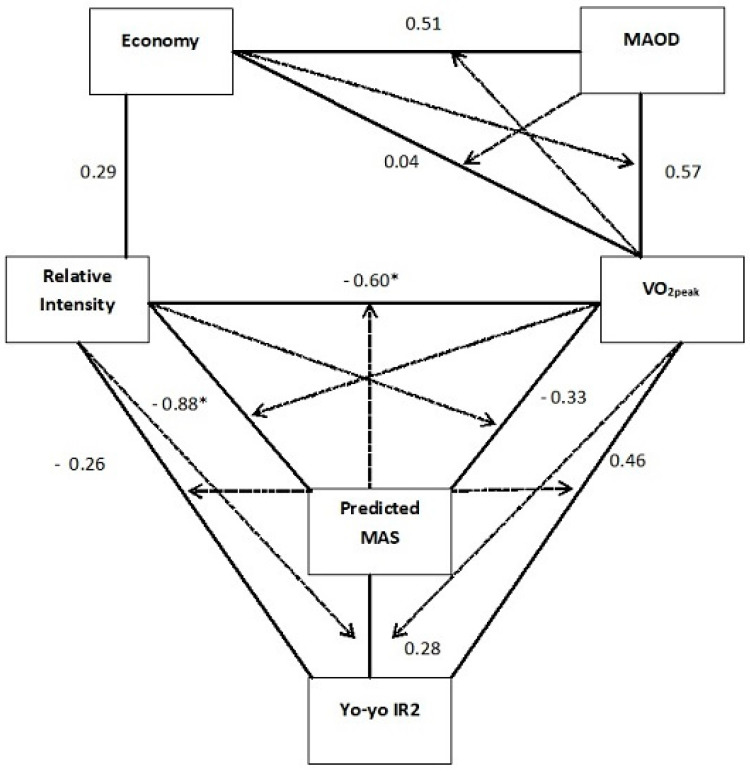
Partial correlation matrix showing the interaction between variables when controlling for other related variables. The solid lines represent the interaction whilst the dotted arrows represent the variable controlling for the interaction. Pearson’s correlation coefficients are represented in this figure with significance at the 0.05 level shown by asterisks.

**Table 1 sports-12-00236-t001:** Correlation matrix between measured variables.

		MAOD	Relative Intensity	Running Economy	Predicted MAS	Yo-Yo IR2 (m)
VO_2peak_ (mL·kg^−1^·min^−1^)	r	0.70	−0.70	0.48	0.54	0.62
	*p*	0.001	0.001	0.04	0.02	0.005
MAOD (mL O_2_ eq·kg^−1^·min^−1^)	r		−0.22	0.66	0.04	0.16
	*p*		0.36	0.002	0.89	0.50
Relative Intensity (%VO_2_ at 12.5 km·h^−1^)	r			0.29	−0.91	−0.72
	*p*			0.22	0.00	0.001
Running Economy (VO_2_ mL·kg·min^−1^ at 12.5 km·h^−1^)	r				−0.40	−0.05
	*p*				0.09	0.86
Predicted MAS (km·h^−1^)	r					0.70
	*p*					0.001

r = Pearson’s correlation coefficients. MAOD = maximal accumulated oxygen deficit. MAS = maximal aerobic speed.

**Table 2 sports-12-00236-t002:** The difference in variables between high and low Yo-Yo IR2 groups (mean ± SD, 95% confidence limits in parentheses).

	High Yo-Yo IR2	Low Yo-Yo IR2	Difference in Mean	*p* Value
VO_2peak_ (mL·kg^−1^·min^−1^)	60.13 ± 4.3	53.93 ± 5.66	6.2 (1.1–11.3)	0.02
MAOD (mL O_2_ eq·kg^−1^·min^−1^)	15.55 ± 6.71	12.43 ± 5.26	3.1 (−2.7–8.9)	0.27
Relative Intensity (%VO_2_ at 12.5 km·h^−1^)	78.93 ± 4.57	87.92 ± 7.83	−9.0 (−15.5–−2.4)	0.01
Running Economy (VO_2_ mL·kg^−1^·min^−1^ at 12.5 km·h^−1^)	47.39 ± 3.43	47.16 ± 4.1	0.22 (−3.5–4.0)	0.9
Predicted MAS (km·h^−1^)	15.84 ± 1.23	14.38 ± 1.24	1.5 (0.2–2.7)	0.02

High (*n* = 8) and low (*n* = 11) Yo-Yo IR2 scores. MAOD = maximal accumulated oxygen deficit. MAS = maximal aerobic speed.

## Data Availability

Requests for data can be forwarded to Brendan O’Brien.

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
