# Peer review of "Anaerobic and Aerobic Metabolic Capacities Contributing to Yo-Yo Intermittent Recovery Level 2 Test Performance in Australian Rules Footballers"

_sports, 2024, doi:10.3390/sports12090236_

Round 1

Reviewer 1 Report

Comments and Suggestions for Authors

One of the claimed strengths of the Yo-Yo IR2 test is its ability to predict repeated sprint performance by challenging a variety of physiological systems in a manner like match play in many team sports, such as soccer. However, Australian football is a unique sport with distinct movement patterns, not widely understood outside of its regional popularity. Therefore, comparing it directly to other sports like soccer may not fully justify the test's applicability. The purpose of this article is to evaluate the Yo-Yo IR2 test as a tool for estimating both aerobic and anaerobic metabolic capacities, with the aim of aiding Australian football coaches in assessing player fitness.

To contextualize the relevance of the Yo-Yo IR2 test for Australian football, it is essential to refer to studies that have examined the sport's specific movement demands, including speed, duration, and acceleration. Data from tracking systems such as Catapult and Polar Team Pro provide insights into the typical movement profiles of Australian football players. These systems have revealed the high-intensity intermittent nature of the sport, with players frequently engaging in short sprints, rapid accelerations, and changes of direction, which align well with the demands measured by the Yo-Yo IR2 test.

Furthermore, the study builds upon existing literature validating the use of maximal accumulated oxygen deficit (MAOD) and VO2max for assessing aerobic and anaerobic capacities. References such as [author, year] and [author, year] have demonstrated the reliability of these measures in estimating maximal aerobic speed (MAS) and their correlation with athletic performance. These validations support the use of these metrics in Australian football for both evaluating fitness and designing targeted training programs.

It is crucial to differentiate the approach used in this study from those focused on running performance. While running performance typically emphasizes continuous aerobic capacity and running economy, this study adopts a more nuanced perspective that considers the intermittent, multi-directional demands of Australian football. The Yo-Yo IR2 test offers a unique advantage by capturing the sport's specific energy system requirements, which include both sustained aerobic effort and rapid anaerobic bursts, thus providing a comprehensive tool for coaches to assess and enhance player performance.

By introducing references that validate the specific movement patterns in Australian football and supporting the use of physiological measures like MAOD and VO2max, this study provides a justified and sport-specific rationale for the application of the Yo-Yo IR2 test, distinct from traditional running assessments.

Author Response

Reviewer 1 comments

Comments 1

One of the claimed strengths of the Yo-Yo IR2 test is its ability to predict repeated sprint performance by challenging a variety of physiological systems in a manner like match play in many team sports, such as soccer. However, Australian football is a unique sport with distinct movement patterns, not widely understood outside of its regional popularity. Therefore, comparing it directly to other sports like soccer may not fully justify the test's applicability. The purpose of this article is to evaluate the Yo-Yo IR2 test as a tool for estimating both aerobic and anaerobic metabolic capacities, with the aim of aiding Australian football coaches in assessing player fitness.

Response 1

We appreciate the reviewer’s comments about the specificity of different sports having different movement requirements that are reflected differently in field test. We outline that research has previously highlighted a mediation relationship between yo-yo IR2 performance and on field exercise intensity and performance metrics in Australian Rules Football. We further highlight that yo-yo IR2 test performance strong correlation to on-field exercise intensity measures in Australian Rules Football are similar to its correlations to on-field performance in soccer. [line 33-35]

Comments 2

To contextualize the relevance of the Yo-Yo IR2 test for Australian football, it is essential to refer to studies that have examined the sport's specific movement demands, including speed, duration, and acceleration. Data from tracking systems such as Catapult and Polar Team Pro provide insights into the typical movement profiles of Australian football players. These systems have revealed the high-intensity intermittent nature of the sport, with players frequently engaging in short sprints, rapid accelerations, and changes of direction, which align well with the demands measured by the Yo-Yo IR2 test.

Response 2

We agree and have referenced the first paper to highlight this in the references [5] Mooney M, O'Brien B, Cormack S et al. The relationship between physical capacity and match performance in elite Australian football: A mediation approach. J Sci Med Sport. 2011;14(5), 447-452. doi: 10.1016/j.jsams.2011.03.010

Comment 3

Furthermore, the study builds upon existing literature validating the use of maximal accumulated oxygen deficit (MAOD) and VO2max for assessing aerobic and anaerobic capacities. References such as [author, year] and [author, year] have demonstrated the reliability of these measures in estimating maximal aerobic speed (MAS) and their correlation with athletic performance. These validations support the use of these metrics in Australian football for both evaluating fitness and designing targeted training programs.It is crucial to differentiate the approach used in this study from those focused on running performance. While running performance typically emphasizes continuous aerobic capacity and running economy, this study adopts a more nuanced perspective that considers the intermittent, multi-directional demands of Australian football. The Yo-Yo IR2 test offers a unique advantage by capturing the sport's specific energy system requirements, which include both sustained aerobic effort and rapid anaerobic bursts, thus providing a comprehensive tool for coaches to assess and enhance player performance.

By introducing references that validate the specific movement patterns in Australian football and supporting the use of physiological measures like MAOD and VO2max, this study provides a justified and sport-specific rationale for the application of the Yo-Yo IR2 test, distinct from traditional running assessments.

Comment 3

Thank-you for providing a view that this study offers unique, justified, and novel information.

Reviewer 2 Report

Comments and Suggestions for Authors

TITLE: Anaerobic and aerobic metabolic capacities contributing to Yo-Yo Intermittent Recovery Level 2 test performance.

Abstract

Lines 11-12: I suggest the authors rephrase this sentence. It is confusing and poorly written.

In addition, avoid excessive use of abbreviations. Make the text simpler and more objective.

Introduction

Lines 53-64: The paragraphs are not connected and are difficult to follow. I suggest the authors revise both paragraphs.

Lines 69-72: Include one or more hypotheses.

Methods

Lines 74-76: This is not a description of the experimental design. We are missing a lot of information on how the study was conducted and what the steps were for data collection.

Lines 78-86: Some pieces of information need to be present or better reported. For example, were the volunteers users of any dietary supplements or pharmacological agents that could affect their somatic or cognitive functions and, indirectly, the studied outcomes? Prior research, for example, has suggested that caffeine ingestion may improve performance in short-term, high-intensity exercise bouts. How were the volunteers recruited? Additionally, were the volunteers habituated with the testing procedures? What were the inclusion and exclusion criteria? Please inform.

Moreover, while some control measures are being implemented in the study, other factors such as the participants' nutrition and hydration levels before engaging in the workout, as well as their behavioral characteristics during recovery, are not clearly documented.

Lines 88-90: How was this controlled?

Lines 104-105: How were these speeds determined?

Were blood lactate concentration analyses not performed?

Lines 156-157: Include a results section.

Discussion

Lines 254-256: Include the limitations.

What are the practical applications of your study?

The manuscript needs to be better structured (e.g., spacing and alignment). The paragraphs do not follow a standard; some are short, others long. Additionally, the tables are poorly formatted.

Author Response

Reviewer  2 comments

Comments and Suggestions for Authors

TITLE: Anaerobic and aerobic metabolic capacities contributing to Yo-Yo Intermittent Recovery Level 2 test performance.

Comment 1

Abstract

Lines 11-12: I suggest the authors rephrase this sentence. It is confusing and poorly written.

In addition, avoid excessive use of abbreviations. Make the text simpler and more objective.

Response 1

We have changed the sentence to read more explicitly and with less abbreviations. It now reads:

Nineteen recreational Australian footballers completed a Yo-Yo IR2 test, and on another day a treadmill peak oxygen uptake (VO2peak) and maximal accumulated oxygen deficit test in a randomised counter-balanced order.

We request to keep the abbreviations for the Yo-Yo IR2 test and VO2 peak test as they are universally used abbreviations in exercise and sport science and are required to keep the abstract word count within the accepted limits.

Comment 2

Introduction

Lines 53-64: The paragraphs are not connected and are difficult to follow. I suggest the authors revise both paragraphs.

Response 2

We agree that the sentence could be improved. We have now linked the paragraphs (reduced from 2 to 1 paragraph) and changed the sentence that links the two former paragraphs.

The Yo-Yo IR2 test performance has substantial aerobic metabolic contribution as it elicits maximum heart rate.

Comment 3

Lines 69-72: Include one or more hypotheses.

Response 3

Response 3

We have now included a hypothesis at the conclusion of the introduction. Line 72-73

Comment 4

Methods

Lines 74-76: This is not a description of the experimental design. We are missing a lot of information on how the study was conducted and what the steps were for data collection.

Response 4

We appreciate the procedures aren’t supplied in detail in this section. To reduce confusion and repetition we have deleted the “Experimental Approach to the Problem” as it appears to add to the confusion rather than augment it. It is important not to be repetitive. We believe the experimental design can be determined by the materials and methods.

Comment 5

Lines 78-86: Some pieces of information need to be present or better reported. For example, were the volunteers users of any dietary supplements or pharmacological agents that could affect their somatic or cognitive functions and, indirectly, the studied outcomes? Prior research, for example, has suggested that caffeine ingestion may improve performance in short-term, high-intensity exercise bouts. How were the volunteers recruited? Additionally, were the volunteers habituated with the testing procedures? What were the inclusion and exclusion criteria? Please inform.

Response 5

A greater level of detail has been provided on the participants inclusion and application to diet. Line 80-83. Further information was provided about the familiarisation sessions that were performed Line 90-99. The section now reads; Nineteen male regional team sport athletes (Australian Rules Footballers) from various teams volunteered to participate in this study through response to flyers placed at football clubs and the University. Participants were included if they were male and had been actively participating in non-professional Australian Rules Football. Participants were asked to maintain a consistent diet in the 24 hours leading into the testing days, including refraining from the use of caffeine.

Comment 6

Moreover, while some control measures are being implemented in the study, other factors such as the participants' nutrition and hydration levels before engaging in the workout, as well as their behavioral characteristics during recovery, are not clearly documented.

Response 6

We have added the sentence to address this recommendation.

Participants were given 35 min of passive rest before completing the supra-maximal run to exhaustion and were encouraged to drink fluids ad libitum and restricted to ingesting less than 20 grams of carbohydrate to avoid possible gastrointestinal distress.

Comment 7

Lines 88-90: How was this controlled?

Response 7

Participants were asked to maintain a consistent diet and to refrain from caffeine for 24 hours prior to the testing sessions.

Comment 8

Lines 104-105: How were these speeds determined?

Response 8

The speeds were pre-determined by the experimenter’s experience with Australian Rules footballers with knowledge of the standard of competition the footballers participated in. All participants could sustain steady state oxygen consumption running at 14.5 km per hour.

Comment 9

Were blood lactate concentration analyses not performed?

Response 9

Blood lactate assessment as not performed or deemed necessary for the studies purpose.  We firmly believe blood lactate assessment does not offer additional insight above the maximal accumulated oxygen deficit to determine anaerobic capacity. However, we acknowledge the fact it was not measured in the study as a limitation, as we acknowledge some readers may perceive this to be a limitation.

Comment 10

Lines 156-157: Include a results section.

Response 10

 Results section has been included Line 166

Comment 11

Discussion

Lines 254-256: Include the limitations.

Response 11

A paragraph has been dedicated to limitations. Line 265-272 

Comment 12

What are the practical applications of your study?

Response 12

We have added in a practical application section.

  1. Practical applications

Repeated high intensity running is an essential component of Australian Rules Football. The Yo-Yo IR2 test is used in Australian Rules Football training to determine the footballer’s capability to perform the repeated high intensity efforts required during a competitive match. The most likely metabolic performance measures to impact Yo-Yo IR2 are VO2peak, predicted maximal aerobic speed and relative intensity, however anaerobic metabolic performance capacities may also impact on performance. Therefore, training should focus on both aerobic and anaerobic metabolic qualities to improve Yo-Yo IR2 test performance.

Comment 13

The manuscript needs to be better structured (e.g., spacing and alignment). The paragraphs do not follow a standard; some are short, others long. Additionally, the tables are poorly formatted.

Response 13

Paragraphs length can vary depending on the context of the point being made. However, we have made several changes to lengths of the paragraphs, which have resulted in a greater consistency in paragraph length.

Reviewer 3 Report

Comments and Suggestions for Authors

Dear Authors

Please, see the attach.

Comments on the Quality of English Language

Pay attention to the wording and to small mistakes in expression and writing... they can be corrected easily

Author Response

Reviewer 3 comments

Comment 1

Anaerobic and aerobic metabolic capacities contributing to Yo-Yo intermittent Recovery Level 2 test performance..... in soccer? - I think the title should refer to the sport for which the study was conducted. In its current form, it could be understood that the relationships determined by the authors would be generally valid for all disciplines, which is not the case 1. Introduction - At L29-30 you discuss "Australian Rules Football (Australian Football)" and at L34 you bring up "Australian Football and Soccer".

Response 1

We agree and have changed the title to ‘in Australian Rules Footballers’.

Comment 2

I think you will have to clarify this idea better (I know there can be two sports disciplines, but the idea in the introduction does not exactly clarify this aspect, related to the idea of this article) - L33

Response 2

We agree and to improve clarity we have added the word rules in between Australian Football to Australian Rules Football in L33.

Comment 3

"The yo-yo intermittent recovery" - I think it is correct "The Yo-Yo Intermittent Recovery" - L36

Response 3

We agree and have made this change.

Comment 4

 reformulate or clarify: "football and soccer" if you distinguish between different disciplines - L41

Response 4

We agree and this has been changed.

Comment 5

"Australian Football league" I think "Australian Football League" would be correct - L57

Response 5

We agree and this has been changed.

Comment 6

even if the term "VO2peak" is known - when it is used for the first time in the manuscript, it must be clarified, detailed and written completely (an article like this can be read by non-specialists and they should understand what the authors are referring to ), later it can be used with abbreviation (but I think the authors know this aspect).

Response 6

We agree this should be clarified. On line 53, where the term is 1st introduced we clarify what the abbreviation stands for peak oxygen uptake (VO2 peak ).

Comment 7

I recommend paying attention to the terms and forms - The purpose is not clearly presented (it is repeated in a different form, towards the end of the manuscript) 2. Materials and Methods - L75-76

Response 7

We can see the confusion created by the different phrasing of the aims: We have changed the wording of the aims at the start of the discussion to:

“This study aimed to identify the aerobic and anaerobic physiological capacities contributing to yo-yo intermittent recovery, level 2 (Yo-Yo IR2) test performance.”

Comment 8

this phrase needs to be a little more developed, detailed, for clarity and ease of understanding (I repeat, it is important for any Journal, as well as other readers, nonspecialists in this aspect, to understand what the authors claim) - L77-86,

Response 8

We have included a sentence to clarify the phase in the methods.

“VO2peak was determined as the highest value recorded in 30 second intervals and reflects the highest aerobic metabolic rate of the individual.”

Comment 9

Subjects section, very vaguely presented; few subjects and the criteria for inclusion (the only clear criterion being 5 hours) and exclusion in the participating group for this research are not clear;

Response 9

We agree, we have addressed this in accordance with the comments from reviewer 2.

Comment 10

- L75-76 is repeated at L89-90

Response 10

Deleted

Comment 11

- "The Yo-Yo IR2 test was completed on a stable indoor surface using the procedures established previously" until L91 the procedure is not described, only that it is applied "in a randomized, counter-balanced order".

Response 11

These have been re-adjusted and included in the procedures section.

Comment 12

 It is NOT described the time at which the ? is completed, before or after lunch,? after physical activities or after rest period ?? after a few minutes of preparing the body for effort or directly? The tests were applied on days of Australian football activity or non-football days of the week?

Response 12

We have addressed these issues particularly through the procedures section Line 98-112

Comment 13

How long does testing take? Etc etc

Response 13

This was included Line 162

Comment 14

 – many details related to the procedure are missing / the procedure is not clear - L103 more details about "motorized treadmill ergometer at 1% gradient (Cosmed, Italy)" - -L122 "Participants ran..." - "run" correction?

Response 14

We have added the details of the treadmill (T200, Cosmed, Italy). We altered the sentence to “The participants ran at five progressively faster speeds for five minutes at each speed (8, 9.5, 11, 12.5 and 14 km∙h-1) without a break between each speed for 25 minutes of accumulated running time.”

Comment 15

  1. Results

- I have not identified this section, it probably starts at L157

Response 15

This has been updated as reviewer 2 suggestions as well.

Comment 16

- L168-172 – the formulation and presentation of the data is not clear. I recommend

Clarification

Response 16

We have added in the title Results to clarify the data presentation. We will address any specific feedback to improve clarity.

Comment 17

  1. Discussion

-L71-72 "this study aims to determine how these metabolic qualities correlate with to YoYo IR2 performance" and at L204-205: "This study aimed to determine the complex

relationships between endurance qualities, MAOD and Yo-Yo IR2 performance. "??? From

my point of view, the authors need to rephrase and clearly delineate what the purpose of

this study is!!!

Response 17

We can see the confusion created by the different phrasing of the aims: We have changed the wording of the aims at the start of the discussion to:“This study aimed to identify the aerobic and anaerobic metabolic performance capacities con-tributing to yo-yo intermittent recovery, level 2 (Yo-Yo IR2) test performance.”.

Comment 18

- L216-243 the discussions are presented in relation to similar studies, but on other

categories of subjects (other ages) and on another level of training (performance athletes,

compared to non-performance athletes in this article) ... at least the references 23-29. I think

that these discussions should be clarified or reformulated, or other research on subjects with

characteristics similar to those involved in this study should be brought into discussion.

- few references to references from specialized literature in the "Discussions" section

Response 18

We respect and understand this view. However, as there is very limited research on the metabolic contributions to the Yo Yo IR2 performance we cannot compare our results to subjects with similar characteristics. This is why there is only few references in the discussion, as there is little research to compared to.

Comment 19

  1. Conclusions

- L262. 265, 268 - "yo-yo IR2" recommend correction "Yo-Yo IR2"

Response 19

Updated to Yo-Yo IR2

Comment 20

- the first conclusion should be presented in relation to the title and without using

abbreviations. There are no abbreviations in the title either. The conclusion should be clear,

synthetic and directly address the title subject to the reader's attention, not appear in the

form "...VO2peak, predicted MAS ....and relative intensity (%VO2peak at submaximal

speed)." (this last form is specific to the results or discussion section, not the conclusion

section)

Response20

Thank-you for your excellent recommendation. The conclusion has been altered accordingly based on your advice.

“This study aimed to determine the aerobic and anaerobic metabolic performance capacities contributing to yo-yo intermittent recovery level 2 performance. The novel analysis of the physiological contributors to the Yo-Yo IR2 identified the most dominant interacting factors to Yo-Yo IR2 performance were peak oxygen uptake, predicted maximal aerobic speed and relative intensity. Importantly these variables showed that independently none of these variables showed a significant correlation with Yo-Yo IR2 performance. Therefore, Yo-Yo IR2 performance requires contribution from multiple factors interacting that can be monitored to determine physical strengths and weaknesses of individual athletes. Therefore, inferring specific physiological characteristics from a Yo-Yo IR2 test in a precise or reliable manor may be unrealistic at the individual level.”

Comment 21

  1. References

- 21 references out of a total of 30 (70%) are older than 13-14 years, some of them older than

20 years !!!!!!!

Response 21

We updated the paper before submission to SPORTs to incorporate all possible relevant contemporary references. The “older” studies are highly relevant to our paper and their publication date or age does not compromise their integrity or relevance to this project. Very little similar research has been conducted in this field over the last 15 years. The references we identified were the most relevant we could find. Essentially there is a limited number of recent references we could use. It is important to emphasize the quality and impact of “old” research relevance is not necessarily diminished by time and importantly, we have used the most relevant references.

Comment 22

- On a topic so specific in sports and so much addressed in literature, I think 30 references

are few.

Response 22

We would add more references if they were genuinely contributed to the paper. This paper is on the short report spectrum of articles size. Adding references to bolster numbers for numbers sake should not occur. There has been little research been completed in the topic and we believe we have accessed all relevant articles. Adding superfluous papers may detract from the aim of the paper.

Comment 23

  1. Technical aspects

- Sports template is not respected

Response 23

We respect the Sports Template. We have attempted to follow the journal’s guidelines and if we have made a mistake in formatting, if we can be directed to the template breaches then we will promptly address and apologize.

Comment 24

- I recommend greater attention to expression, to the formulation of ideas

Response 24

We have made significant changes to the document to improve our expression and formulation of idea based on the 3 reviewer’s comments.

Comment 25

- I notice that data from 2011 is used (L278 project approval number A11-002, 2011)." - I

think that the authors should also clarify this aspect, because the tests, I think, were applied

in 2011 and then I ask: aren't the data from a social and professional context in which the

activities had a different specificity than the current one? Especially in the conduct of

Australian football / soccer activities?!!

Response 25

The data was collected in 2011, however the game of Australian Football has not changed as the game duration is still four 20-minute quarters (80 minutes) with the same rest periods between quarters. The movement patterns required in the game have not changed. There have been some very minor changes in rules, however this has not affected the game movement requirements and need for repeated high intensity efforts. The yo-yo IR2 remains as relevant test today as it did 15 years ago, and the professional/social context would not have altered the test relevance.

Round 2

Reviewer 1 Report

Comments and Suggestions for Authors

Your responses validate that no modifications have been made to the manuscript.

Nevertheless, I believe you may be able to offer valuable insights into this Australian sport, which could contribute to its increased visibility on a global scale, potentially leading to its inclusion in future Olympic Games. 😉

Reviewer 2 Report

Comments and Suggestions for Authors

The authors have carefully reviewed and incorporated my comments.

Reviewer 3 Report

Comments and Suggestions for Authors

Thanks for the answers. I hope that the editor's decision is an academic one.

I remain of the opinion that some sources, regardless of how well-known their authors are, are still too old to make a real contribution to the novelty of an article that the authors want to be original and new

I wish you success